# New Insights into the Link between Melanoma and Thyroid Cancer: Role of Nucleocytoplasmic Trafficking

**DOI:** 10.3390/cells10020367

**Published:** 2021-02-10

**Authors:** Mourad Zerfaoui, Titilope Modupe Dokunmu, Eman Ali Toraih, Bashir M. Rezk, Zakaria Y. Abd Elmageed, Emad Kandil

**Affiliations:** 1Department of Surgery, Tulane University School of Medicine, 1430 Tulane Avenue, SL-22 New Orleans, LA 70112, USA; titilopedokunmu@gmail.com (T.M.D.); etoraih@tulane.edu (E.A.T.); ekandil@tulane.edu (E.K.); 2Department of Biochemistry, Covenant University, Ota 23401, Nigeria; 3Department of Natural Sciences, Biology Unit, Southern University at New Orleans, New Orleans, LA 70126, USA; batteia@suno.edu; 4Department of Pharmacology, Edward Via College of Osteopathic Medicine, University of Louisiana at Monroe, Monroe, LA 71203, USA; zelmageed@ulm.vcom.edu

**Keywords:** nucleocytoplasmic transport, tumor aggressiveness, resistance, BRAF V600E, thyroid cancer, melanoma

## Abstract

Cancer remains a major public health concern, mainly because of the incompletely understood dynamics of molecular mechanisms for progression and resistance to treatments. The link between melanoma and thyroid cancer (TC) has been noted in numerous patients. Nucleocytoplasmic transport of oncogenes and tumor suppressor proteins is a common mechanism in melanoma and TC that promotes tumorigenesis and tumor aggressiveness. However, this mechanism remains poorly understood. Papillary TC (PTC) patients have a 1.8-fold higher risk for developing cutaneous malignant melanoma than healthy patients. Our group and others showed that patients with melanoma have a 2.15 to 2.3-fold increased risk of being diagnosed with PTC. The *BRAF V600E* mutation has been reported as a biological marker for aggressiveness and a potential genetic link between malignant melanoma and TC. The main mechanistic factor in the connection between these two cancer types is the alteration of the RAS-RAF-MEK-ERK signaling pathway activation and translocation. The mechanisms of nucleocytoplasmic trafficking associated with RAS, RAF, and Wnt signaling pathways in melanoma and TC are reviewed. In addition, we discuss the roles of tumor suppressor proteins such as p53, p27, forkhead O transcription factors (FOXO), and NF-_K_B within the nuclear and cytoplasmic cellular compartments and their association with tumor aggressiveness. A meticulous English-language literature analysis was performed using the PubMed Central database. Search parameters included articles published up to 2021 with keyword search terms melanoma and thyroid cancer, BRAF mutation, and nucleocytoplasmic transport in cancer.

## 1. The Connection between Melanoma and Thyroid Cancer: Our Up-to-date Knowledge

Several studies, including ours, have observed that patients with malignant melanoma have a higher risk of developing other primary cancers, including thyroid cancer (TC) [1,2,3]. Patients with melanoma have a 2.3-fold higher risk of being diagnosed with papillary TC (PTC), and patients with PTC have a 1.8-fold higher risk of developing melanoma [1]. Our analysis showed a two-fold increased risk of second primary TC (SPTC) following melanoma (The Standardized Incidence Ratio (SIR) = 2.15, 95% confidence interval (CI) = 1.99–2.32) compared with the general population (Figure 1A). A significantly elevated risk of SPTC was noted in the first year of melanoma diagnosis: SIR = 5.42 (95% CI = 4.65–6.28). Persistently increased risk of SPTC was evident beyond the first year of follow-up: within five years SIR = 2.05 (95% CI = 1.81–2.32) and within 10 years SIR = 1.54 (95% CI = 1.31–1.81) (Figure 1B).

There may be a genetic link between cutaneous melanoma and TC due to the high occurrence of a mutation in the BRAF oncogene. Tissue specimens from patients with melanoma and/or TC show a high rate of the *BRAF V600E* mutation. *BRAF* mutations are observed in 36 to 83% of cases of PTC in all age groups [4] and in 63% of melanoma cases [5]. Dysregulation of the RAS-RAF-ERK pathway and thus persistent nuclear translocation of ERK and other signaling molecules is a major common cause in the development of melanoma and TC. Other common genetic alterations associating TC with melanoma include the receptor tyrosine kinase *RET* (rearranged during transfection) gene mutation and *RAS*. Research shows an altered *RET* gene is present in 10 to 30% of PTC and RET rearrangements have been reported in melanomas [6].

## 2. Dysregulation of the Nucleocytoplasmic Trafficking

Nucleocytoplasmic trafficking is the transport of proteins, RNAs and signaling molecules between the nucleus and the cytoplasm [7]. Generally, about 50% of proteins are transported across the nucleus and cytoplasm or other cellular compartments to reach their site of function [8]. Intracellular trafficking regulates biochemical activities such as gene expression in eukaryotes. DNA synthesis and transcription take place in the nucleus, but mRNA must be transferred to the ribosome in the cytoplasm for translation. Similarly, proteins and signaling molecules shuttling between the nucleus and other compartments require transporter-complexes, such as karyopherins/importins, in normal and cancer cells [7,8,9,10,11,12,13,14,15,16,17,18]. Nucleocytoplasmic transport of proteins is achieved when nuclear localization sequences (NLS) and nuclear export sequences (NES) on the cargo protein form a complex with importin or exportin, and the cargo-receptor complex then bind to nucleoporins via the receptor. The widely studied family of nuclear transporters include importins (Importin-β2, importins α and β) and exportins (chromosome region maintenance or CRM1) [9,11,12,15,19,20]. 

CRM1 is the main exporter from the nucleus of tumor suppressor and other cargo proteins in eukaryotic cells. It exports proteins by binding to small Ran GTPase to actively transport the proteins across the nuclear membrane [12,19,21,22,23,24,25,26]. Inhibition of the activity of CRM1 has been extensively explored as a therapeutic target to inhibit shuttling nucleocytoplasmic transport in melanoma, thyroid, and other types of cancers [9,11,12,17,19,23,24,26,27,28,29,30,31,32]. In Table 1, we summarize the members of the importins/exportins family with some examples of their cargoes. In certain situations, aberrant nucleocytoplasmic trafficking has been implicated in different types of diseases including cancers such as thyroid and melanoma [8,13,15].

Tumorigenesis involves complex alterations in tumor suppressor genes and activation of oncogenes from proto-oncogenes, which promote growth signaling pathways to induce neoplastic transformation in normal cells [33]. The translocation and activation of oncogenes initiates tumorigenesis, cell growth, and resistance to chemotherapeutic drugs [8,9,11,13,25,32,34,35,36]. These oncogenes implicated in tumorigenesis undergo unrepaired DNA damage that results in mutations that initiate and/or promote tumorigenesis [8,37,38,39]. DNA damage results in mutations or impaired gene functions that alter post-translational modifications or disrupt the regulatory network of cells, leading to uncontrollable cell growth. Oncogenes and tumor suppressor proteins localized in the nucleus play critical roles in cancer development. They have been targeted for anticancer therapy, and a number of them are common between melanoma and TC.

## 3. Dysregulation of Nucleocytoplasmic Transport in Melanoma and Thyroid Cancer

Emerging evidence indicates a rising incidence of melanoma and TC in the United States, with reports of over 100,350 and 52,890 cases in 2020, respectively, in the Surveillance, Epidemiology, and End Results (SEER). To understand then link between melanoma and TC aggression, we reviewed nucleocytoplasmic transport and associated molecular mechanisms common to both cancers. 

Aggressive forms of melanoma are associated with an increased risk of developing thyroid malignancies [40], possibly via the expression of thyroid-stimulating hormone (TSH), which converts melanocytes to melanoma. Since the TSH is elevated in patients with thyroid failure and the TSH receptors are highly expressed in melanomas, it has been postulated that TSH activates the TSHR signaling pathways, which are critical in the development of melanoma [41,42]. Immune checkpoint inhibitors such as PD-(L)1 blockade in melanoma can also trigger a thyroid dysfunction. Pathological associations exist between melanoma and TC, and this is stirring up interest in understanding mechanisms common to both cancers and how resistance to BRAF inhibitors treatment has common mechanisms of survival [41,43]. 

Mechanisms of nucleocytoplasmic transport can be importin-dependent, export-dependent, or unaided. Endogenous transcriptional activator-nuclear factor-κB (NF-κB), extracellular signal-regulated kinase 2 (ERK2), β-catenin, and p53 accumulate in the cytoplasm in the basal state, where they interact with other signaling partners that restrict them to the cytoplasm. NF-κB plays important roles in cell proliferation, immune response and inhibition of apoptosis and is associated with resistance to anticancer therapies. It forms a complex with IκB, which is an inhibitor of NF-κB import into the nucleus. In cancer cells, IκB is phosphorylated by the IΚΚ complex, causing IκB degradation and nuclear import of NF-κB. Dysregulation of NF-κB nucleocytoplasmic transport leads to the promotion of tumorigenesis in TC [14,44] and to BET inhibitors (Bromodomain and extra terminal protein inhibitors) resistance in melanoma [45]. In melanoma and TC, unidentified importin and CRM1 shuttle RAF proteins (particularly BRAF) into and out of the nucleus, which phosphorylate MAPK/ERK kinases (MEKs) within the mitogen-activated protein kinase (MAPK) cascade upon activation by proto-oncogenes Rat sarcoma (RAS) proteins [46]. 

Activation of RAS proteins, including *HRAS*, *KRAS*, and *NRAS* genes, activate growth signaling factors like those mediated by tyrosine kinase receptors including MAP kinase, Phosphoinositide 3-kinase (PI3K), and protein kinase B (also called Akt). Together, they promote cell proliferation and survival [39] under healthy conditions, but in tumorigenesis, they mediate progression and aggressiveness [38,47,48,49,50,51]. 

MAP kinase, PI3K, and V600E mutation in BRAF oncogene are common features in melanoma and TC [52,53]. On the cellular level, RAS and BRAF mutations, but not ARAF or CRAF mutations, play significant roles in the ERK signaling pathway [54,55,56]. For example, ERK accumulates in the nucleus following stimulation by mitogenic signals. Nuclear export of ERK is inhibited by blocking CRM1 because the relocalization of nuclear ERK to the cytoplasm involves MEK1, which contains the NES sequence [57]. Other key signaling molecules, such as forkhead O transcription factors (FOXO), p27, β-catenin, p53, and claudin-1 are trafficked in melanoma and TC cells and are associated with cell proliferation and progression.

FOXO transcription factors (FOXO1a, FOXO3a and FOXO4) are negative regulators of cell proliferation, survival and progression [58]; they are inactivated through phosphorylation by Akt, which promotes its nuclear export [59,60,61]. Nuclear localization of Akt in thyroid cells increases oncogenic expression, is associated with high metastatic invasion in lymph nodes, and plays a significant role in tumor aggression [62,63,64,65,66,67], which could be dependent on p27 tumor suppressor gene cytosolic levels. Under normal conditions, the cell-cycle inhibitor p27 is localized in the nucleus where it binds to and inhibits Cyclin-dependent kinase 2 (CDK2); in many carcinomas including TC, p27 localization is mainly cytoplasmic, resulting in cell-cycle progression and tumorigenesis. Active PI3K signaling leads to activated Akt and phosphorylation at the NLS of p27 and its subsequent cytoplasmic sequestration [14]. 

In the Wnt signaling pathway, β-catenin (*CTNNB1* gene) activates T-cell factor- or lymphoid enhancer factor-regulated gene transcription. It is regulated by glycogen synthase kinase 3β (GSK-3β) phosphorylation of serine and threonine residues, which stabilizes β-catenin and prevents its degradation in the cytoplasm [14,68]. In melanoma and thyroid cancer, β-catenin is bound to cadherins and α-catenin, which form cell-cell adhesion complexes that inhibit the nuclear import of β-catenin [69]. β-catenin plays a key role in the cadherin/catenin complex involved in cell–cell adhesion, the loss of which may lead to tumor invasion and metastasis [70]. β-catenin can be transported by itself, through the aid of CRM1, or translocated by binding to adenomatous polyposis coli (APC) and axin. APC and axin can be trafficked to the nucleus and bind to nuclear β-catenin for export from the nucleus, both of which involve NES for CRM1-mediated export [71]. Mutations in the GSK-3β phosphorylation sites are rare but result in cellular accumulation of β-catenin. 

In melanoma, activation of the Wnt signaling pathway promotes migration, invasion and proliferation, linking Wnt signaling to more aggressive behavior and worse prognosis [72,73,74]. Primary and metastatic tumors display widespread cytoplasmic β-catenin and the loss of nuclear expression of β-catenin has been associated with cancer progression. Wnt5A is a known ligand in the non-canonical Wnt pathway, which inhibits β-catenin, a key process in the canonical Wnt pathway. Increased expression of cytoplasmic Wnt5A has been associated with melanoma progression and poor outcomes [75]. However, increased nuclear β-catenin correlates with reduced proliferation and tumor size and improves survival in malignant tumors [72].

p53 is a tumor suppressor protein that is activated in response to DNA damage and other stresses. It accumulates in the nucleus where it mediates cell-cycle arrest, apoptosis, and senescence. Its activation involves phosphorylation and acetylation leading to subcellular mislocalization [76,77,78,79]. The nucleocytoplasmic export of p53 is mediated by CRM1. Most tumors have mutated p53 with low frequencies in melanoma, and cytoplasmic accumulation of wild-type p53 is reported in breast and colorectal carcinoma. Nuclear accumulation of p53 protein is associated with the de-differentiation of papillary carcinoma [80]. p53 nucleocytoplasmic transport has been targeted by nuclear transport inhibitors as a potential mechanism to inhibit tumor aggressiveness [81]. 

Another protein trafficking dysregulation in melanoma and TC is related to claudin-1: Its cytoplasmic localization is reported in invasive forms of melanoma, whereas claudin-1 nuclear localization is found in benign nevi [82,83]. Claudin-1 proteins are important in the formation of tight junctions, which promote adhesion and growth and enhance transport of molecules across the cell membrane. In one interesting in vitro study, melanoma cells transfected with NLS-claudin-1 vector showed significant nuclear localization of claudin-1, but still had transport of claudin-1 to the cytoplasm. This translocation can be controlled by phospholipase A (PKA) phosphorylation and can affect metastatic capacity [82]. In TC, Zwanziger et al. showed increased nuclear claudin-1 localization in follicular TC metastases [84].

Table 2 shows the nucleocytoplasmic transport dysregulation of tumor-suppressor proteins, transcription factors and signaling molecules such as FOXO, p53, and NF-κB in melanoma and TC [36,49,62,64,66,85,86,87,88,89].

## 4. Nucleocytoplasmic Transport and Mechanisms of Resistance in Cancer

Multiple factors are involved in progressive and aggressive melanoma and TC with overlapping resistance mechanisms [48,53,66,90] as depicted in Figure 2A.

Acquired resistance to BRAF kinase inhibitors is mediated by reactivation of MAPK signaling, which elevates ERK1/2 phosphorylation and translocation to the nucleus. Carlino and colleagues detected a maintained phospho-ERK expression in all resistant sublines in the presence or absence of BRAF and/or MEK inhibitors [46,91].

Another resistance mechanism is mediated by p53. Wnt5A is a non-canonical Wnt ligand that drives a metastatic, therapy-resistant phenotype in melanoma. It increases the half-life of wild-type nuclear p53 to promote a slow-cycling phenotype while inhibiting p53-induced apoptosis via increased iASPP (inhibitor of apoptosis-stimulating protein p53) activation and translocation to the nucleus. Inhibitors of p53 block the slow-cycling phenotype and sensitize melanoma cells to BRAF/MEK inhibitors [92]. 

In addition to ERK, and p53, the overexpression of CRM1 has been linked with poor prognosis and resistance to treatment in melanoma [93,94]. Inhibiting aberrant translocation of proteins between nucleus and cytoplasm has shown high therapeutic advantage in many different cancers including melanoma and TC [35,93]. One study has shown synergistic effects of CRM1 and BRAF inhibitor combinations with effective tumor regression in BRAF-mutant melanoma [95]. CRM1 may be associated with drug resistance in several cancers by nuclear export of drug targets, including topoisomerase IIα, Bcr-Abl, and Galectin-3 [96]:Galectin-3: It interacts with a wide range of partners and has multiple activities in cancer cells. Subcellular localization of Galectin-3 is important for its function as a regulator of apoptosis [97]. Phosphorylated cytoplasmic Galectin-3 activates ERK and c-Jun N-terminal kinase (JNK), resulting in subsequent suppression of apoptosis in cancerous cells. Treatment with cisplatin, a pro-apoptotic agent, can lead to movement of Galectin-3 to the cytoplasm, resulting in drug resistance. CRM1 inhibition by leptomycin B prevents nuclear export of Galectin-3 and restores cisplatin-induced apoptosis in cancer cells [98].Topoisomerase IIα: Cancer cells can develop drug resistance to the cytotoxic effects of topoisomerase II inhibitors like doxorubicin by exporting topoisomerase IIα from the nucleus to the cytoplasm by a CRM1-mediated mechanism. Topoisomerase IIα participates in DNA replication and transcription. Doxorubicin targets topoisomerase IIα, producing DNA-cleavable complexes and cell death. For DNA damage to occur, topoisomerase IIα must be localized in the nucleus. CRM1 inhibition can block the nuclear export of topoisomerase IIα and sensitize cancer cells to treatment with doxorubicin [99]. Bcr-Abl: The chromosomal translocation between chromosomes 9 and 22 leads to the formation of a new gene called *Bcr-Abl*. This gene produces the tyrosine kinase Bcr-Abl protein, which is localized in the cytoplasm where it activates proliferative and anti-apoptotic signaling pathways. However, the presence of Bcr-Abl kinase protein in the nucleus followed by its activation along with p73 will result in DNA damage-induced apoptosis. Targeting of Bcr-Abl kinase by imatinib in combination with leptomycin B leads to nuclear retention of Bcr-Abl kinase and promotes apoptosis in imatinib-resistant chronic myeloid leukemia (CML) cells [100]. 

One example of the non-common mechanism of tumor aggressiveness is the dysregulated transport of thyroid transcription factor 1 (TTF-1). In PTC, reduced nuclear localization of TTF-1 is linked to vascular invasion and nodal metastases and is a strong predictor of tumor recurrence in the presence of BRAF mutation [101].

## 5. Targeting Nucleocytoplasmic Transport

The high pathophysiological relevance of importins and exportins highlights their potential as therapeutic targets for melanoma and TC.

### 5.1. Targeting Nuclear Import

Currently, the development and use of protein nuclear import inhibitors for cancer treatment lags behind that of nuclear export inhibitors, and the former have not yet entered clinical trials. The first nuclear import inhibitor was developed in 1995 by Lin et al., who found that cell-permeable peptides were inhibiting nuclear translocation of NF-κB in intact cells [102]. In 2010, Ambrus et al. [103] used a novel screening approach to identify small molecule inhibitors of the importin α/β pathway such as 58H5-6. However, since no inhibitory effects could be observed in vivo, the 58H5-6 inhibitor could not move forward toward clinical trials. In the following years, potent inhibitors of importin α/β-mediated nuclear import were identified: M9M by Cansizoglu [104], karyostain 1A by Hintersteiner et al. [105], importazole by Soderholm et al. in 2011 [106] and INI-43 by Van der Watt in 2016 [107]. Although some of these inhibitors show high potential, their therapeutic applicability has not yet been investigated. Further research is required to establish import nuclear transport inhibitors as a therapeutic intervention in melanoma and TC.

### 5.2. Targeting Nuclear Export

CRM1 is the major exporter of proteins from the nucleus to the cytoplasm, and for several years it has been the only exportin targeted by inhibitors. Leptomycin B (LMB), a specific CRM1 inhibitor, has been characterized [108] and its role has been widely described [109,110,111]. However, due to toxicity, phase I clinical trials with LMB were stopped [112]. Consequently, several LMB analogs were developed, such as ratjadones, groniothalamin, and KOS-2464. For various reasons, none made it to clinical trials [113,114,115], but novel synthetic CRM1 inhibitors such as CBS9106 (SL-801) moved to clinical trials [116]. A new class of selective inhibitors of nuclear export (SINE) including KPT-185, KPT-251, KPT-276, KPT-330 (Selinexor) and KPT-335 (Verdinexor) are extremely selective and used as anti-cancer agents [117,118]. Some of the SINE compounds are currently being tested in phase I/II/III clinical trials to treat solid organ malignancies, as single agents and in combination with standard therapies.

## 6. Conclusions

We found that cutaneous malignant melanoma increases the risk of papillary TC and vice versa. In addition, patients with both cancers have a high frequency of *BRAF V600E* mutation. Clinical outcomes following treatment with one class of drugs (such as BRAF or MEK/ERK inhibitors) for TC and melanoma are not impressive. Hence, cancer therapy targeting multiple pathways with combinations of SINE, MEK/ERK, PI3K, and kinase inhibitors have been developed for the treatment of melanoma and TC, several of which show improved responses including sorafenib, lenvatinib, and others [47,48,49,50,51,119,120]. Since effective cellular functioning relies on active/passive transport of molecules to other compartments or localization in a specific site, SINE such as CRM1 antagonists, which block nuclear export or alter post-translational modification of cargo proteins, are being tested in combination with different drugs [10,19,26,27,28,121]. Phase II clinical trials of SINE inhibitors such as selinexor, sorafenib and other kinase inhibitor combination therapies with other classes of antineoplastic drugs show effectiveness in reducing tumorigenesis [96,119]. SINE inhibitors of nuclear export mechanisms decrease tumor progression and invasiveness [14,46].

## Figures and Tables

**Figure 1 cells-10-00367-f001:**
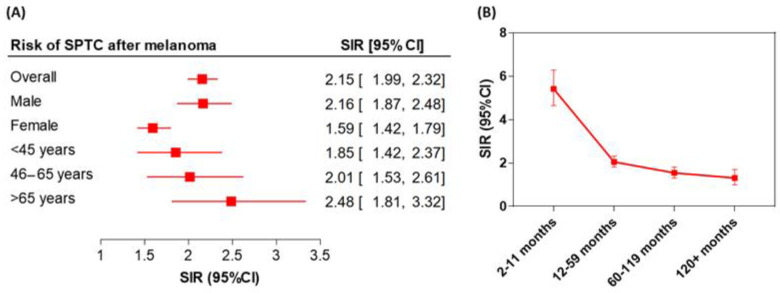
Risk of second primary thyroid cancer (SPTC) following malignant melanoma. (**A**) Standardized Incidence Ratio (SIR) stratified according to the gender and age groups. (**B**) Latency course of risk of SPTC within the first 10 years of follow-up of melanoma patients. The error bars indicate the 95%CI.

**Figure 2 cells-10-00367-f002:**
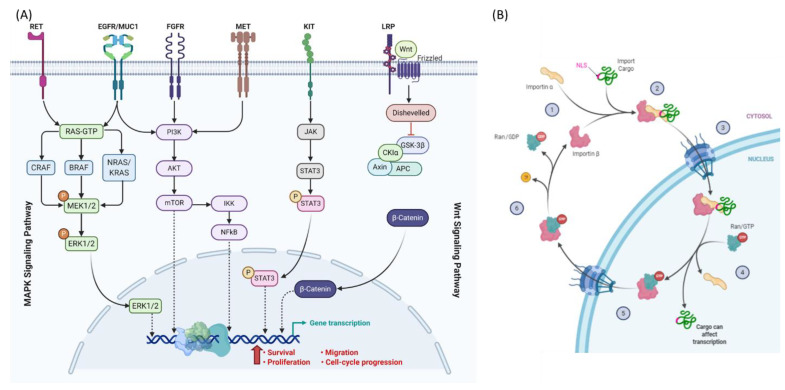
Nucleocytoplasmic translocation. (**A**) Signaling molecules trafficking in the MAPK and WNT signaling pathways in melanoma and thyroid cancer. (**B**) Example for the transport cycle of importins. Import of NLS-containing cargos is mediated by importin αβ. Importin α recognizes and binds the NLS of the cargo as well as importin β. After transport through the nuclear pore channels (NPC), the GTP-bound GTPase Ran binds to importin β and dissociates the import complex. Next, Ran/GTP transports the complex across the NPC into the cytoplasm. Hydrolysis of GTP to GDP is stimulated by Ran binding protein 1 and Ran GTPase activating protein 1 in the cytoplasm. Conformational rearrangements in Ran/GDP lead to the dissociation of importin β to be free for the next cycle. Created with BioRender.com.

**Table 1 cells-10-00367-t001:** Nucleocytoplasmic transport factors with examples of their cargoes.

Transport Factor	Cargoes
***Exportins***	
Exportin-1 (Crm1)	Leucin-rich NES cargoes, NF-kB, Cyclin D1, NFAT, p53, p21, IkB, BCR-ABL, FOXO3a, TOPO IIa, eIF4E, HIV genomic RNA
Cellular apoptosis susceptibility (CAS/XPO2)	Importin alpha
Exportin-t	tRNA
Exportin 5	Pre-microRNA, tRNA, eEF-1A, ILF3, Staufen2, dsRNA-binding proteins, 60S pre-ribosomal subunits
Exportin 6	Profilin, Actin
Exportin 7	P50Rho-GAP, Histone 2A, Histone H3, 14-3-3
***Importins***	
Importin β1	Cargos with basic NLs via importin alpha, NFAT, PRPF31, CREB, p65, β-catenin, JAK1, STAT5, cyclin B1, SRY/SOX-9, PTHrP
Importin β2	Histone, ribosomal proteins, FOXO4, FUS, hnRNAPA1
Importin β3	c-Jun, Histones, ribosomal proteins, IRF3, RASAL2, HPV E5 (16E2)
Importin 3	HuR
Importin 4	HIF1-alpha, Histones, ribosomal proteins, Vitamin D receptor
Importin 7	c-Jun, CREB, Ribosomal proteins, SMAD3, HIV RTC, GR, Histone H1
Importin 8	SMADs, eIF4E, Signal Recognition Particle Protein 19
Importin 9	c-Jun, PP2A (PR65), NUAK1, nuclear actin, Histone, ribosomal proteins,
Importin 11	UBE2E3, UBE2E1, PTEN, β-catenin, UBcM2, rpL12
Importin 12	SRSF1, CIRBP
***Import/Export***	
Importin 13	Import: c-Jun, Mago-Y14, RBM8, Ubc9, Glucocorticoid Receptor, Pax6 Export: eIF1A
Exportin 4	Import: Sox2, SRY Export: SMAD3, eIF5A
***Non-characterized***	
Ran BP6	Undefined
Ran BP17	Undefined

**Table 2 cells-10-00367-t002:** Nucleocytoplasmic mechanisms of aggressive melanoma and thyroid cancer.

Signal Transducer	Translocation Effects	Oncogenic Role	Specific Cancer	References
FOXO1, FOXO3a, FOXO4, FOXO6	Cytoplasmic mislocalization promoted by Akt. Nuclear localization of Akt in thyroid cells increases oncogenic expression, high metastatic invasion in lymph nodes and tumor aggression	Activate transcription of genes that triggers cellular proliferative, cell cycle, differentiation, and cell death.	Melanoma, thyroid cancer	Kau et al., 2004; Tang et al., 1999, Takaishi et al., 1999; Nakamura et al., 2000
Claudin-1	Translocation from nucleus to cytoplasm in melanoma cells and increased cytoplasmic expression in a PKC-dependent manner but altered migration by PKA Phosphorylation.	Increased expression, invasiveness in melanoma hence a marker of progression	Melanoma	French et al., 2009; Leotlela et al., 2007
B-catenin	Nuclear expression	Tumor suppressor role in primary and secondary tumors	Melanoma, thyroid cancer	Chien et al., 2009
Cyclin D1	Cytoplasmic claudin-1 is highly expressed with more aggression and increased invasiveness in melanoma unlike benign nuclear claudin-1	Accumulation of cells in the G1 phase of cell cycle.	Melanoma	French et al., 2009; Leotlela et al., 2007
CDKN1B (p27)	Phosphorylated by Akt and exported from nucleus to cytoplasm. Cytoplasmic expression is associated with poor 5-year survival in metastatic melanoma	A cell-cycle inhibitor, blocks cell cycle in the G0/G1 differentiation signals or cellular stress —cell cycle, activation of PI3K and MEK-dependent kinases	Thyroid, melanoma	Kau et al., 2004
p53	Mutation, post-translational modification, or cytoplasmic mislocalization	Acts as a tumor suppressor and trigger cell cycle arrest, apoptosis, senescence, DNA repair, DNA damage and change the metabolism depending on physiological conditions. Also, known as Guardian of the genome.	Melanoma	Fabbro & Henderson, 2003; Webster et al., 2019
NF-kB	Nuclear import of NF-κB leads to increased target gene expression leading to promotion of tumorigenesis and resistance to anticancer therapies	Activate NF-kB signaling and induce apoptosis of cancer cells.	Thyroid cancer	Kau et al., 2004
Muc 1/EGFR	MUC1 confers survival advantage in melanoma, overexpression of EGFR and nuclear mislocalization is associated with aggressiveness	Induce oncogene expression through interaction with β-catenin and EGFR.	Melanoma and thyroid cancer	Zhao et al., 2014; Patel et al., 2005; Ward et al., 2007

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
