# Peer review of "New Insights into the Link between Melanoma and Thyroid Cancer: Role of Nucleocytoplasmic Trafficking"

_cells, 2021, doi:10.3390/cells10020367_

Round 1

Reviewer 1 Report

Zerfaoui et al summarise in their article current knowledge concerning the role of altered nucleocytoplasmic transport in melanoma and thyroid cancer. This is certainly an interesting topic, but the manuscript in its current state is not suitable for publication.

  1. Overall, the manuscript is lacking logic. In particular the sections "Nucleocytoplasmic transport in melanoma and thyroid cancer" and "Nucleocytoplasmic transport and mechanisms of resistance in cancer" are reading like lists without clear structure and connections between the individual paragraphs. Please make use of connector and avoid one sentence paragraphs.
  2. The figures and tables should be revised to reach an appropriate standard (size, style, font size, figure legend). Is Figure 2 self-made or adapted from another publication?
  3. The authors should make sure to explain abbreviations when using the first time and only then.
  4. Consistent writing of protein names should be used: the authors use Crm1, XPO1, CRM1, all in a tumble. Also Wnt5A and WNT5A.
  5. Line 67/68: the authors have written that nucleocytoplasmic transport is achieved by nucleoporin recognition of NLS and NES sequences. This is incorrect: NLS and NES sequences are recognised by the importins/exportins, and the cargo-receptor complex then bind to nucleoporins via the receptor.
  6. Line 74: "binding of small Ran GTPases ": there is only one, at least in human.
  7. Table 1: please use consistent writing for importin alpha, not greek letters and full word mixed. Also, in line 71, the authors introduce transportin 1, whereas in Table 1 importin beta2 is used as name, please be consistent.
  8. Line 100: Please introduce SEER.
  9. Line 104: what is meant by chaperone-export dependent transport?
  10. Line 124-126: please rephrase the sentence.  
  11. Line 127: Sentence is incomplete.
  12. Line 128: "In melanoma and TC, importin and exportin shuttle RAF proteins...": which importin and which exportin? 
  13. Line 149-151: Transition between the two sentences is unclear.
  14. Line 166: "The nucleocytoplasmic transport of p53 is mediated by CRM1." Do you mean the nuclear export or do you mean that CRM1 regulates the localization of p53. CRM1 for sure does not mediate nuclear import of p53.
  15. Line 172: "claudin-1 nuclear expression..": do you mean localisation? Or please specify.
  16. Line 180-183: please rephrase the sentence.
  17. Line 203: "Inhibiting importin and exportin...": which one?
  18. Line 203-205: the inhibitors do not inhibit the receptors "through phosphorylation of RanGDP to RanGTP". This is wrong in two respects: first, the inhibitors typically bind to the NES/NLS-binding pocket of the receptor, thereby preventing import/export complex formation. Second, the conversion of RanGDP to RanGTP does not occur by phosphorylation, but by an exchange.
  19. Line 226: "Swapping of DNA...": what do you mean by swapping? To me this is an inappropriate word to describe chromosomal translocations.
  20. Line 257: "clinical trials with CRM1...": Do you mean LMB?
  21. Line 259/260: the sentence is incomplete.
  22. Line 280: "SINE inhibitors of nuclear import and export mechanisms...": SINEs do not inhibit nuclear import, as it is in their name: selective inhibitor of nuclear export, they target CRM1exclusively.

Author Response

Reviewer 1:

  1. Overall, the manuscript is lacking logic. In particular the sections "Nucleocytoplasmic transport in melanoma and thyroid cancer" and "Nucleocytoplasmic transport and mechanisms of resistance in cancer" are reading like lists without clear structure and connections between the individual paragraphs. Please make use of connector and avoid one sentence paragraphs. reply: With the extensive revision to the manuscript, in particular to the sections III (line 96) and IV (Line 199), we made sure that the review flows smoothly and logically. Please see all the changes with the “track changes”
  2. The figures and tables should be revised to reach an appropriate standard (size, style, font size, figure legend). Is Figure 2 self-made or adapted from another publication? reply:The 2 figures and 2 table were revised to reach an appropriate standard.
  3. The authors should make sure to explain abbreviations when using the first time and only then.reply:All abbreviations are now only explained when used the first time
  4. Consistent writing of protein names should be used: the authors use Crm1, XPO1, CRM1, all in a tumble. Also Wnt5A and WNT5A.reply:Consistent writing of protein names is checked
  5. Line 67/68: the authors have written that nucleocytoplasmic transport is achieved by nucleoporin recognition of NLS and NES sequences. This is incorrect: NLS and NES sequences are recognised by the importins/exportins, and the cargo-receptor complex then bind to nucleoporins via the receptor. reply: The sentence in line 67/68 was corrected. Now in line 67-69
  6. Line 74: "binding of small Ran GTPases ": there is only one, at least in human. reply: Ran GTPases in now Ran GTPase (Line73)
  7. Table 1: please use consistent writing for importin alpha, not greek letters and full word mixed. Also, in line 71, the authors introduce transportin 1, whereas in Table 1 importin beta2 is used as name, please be consistent. reply: Transportin 1 is now Importin-β2 in the text (Line 70)
  8. Line 100: Please introduce SEER. reply: SEER is introduced (Line 98/99)
  9. Line 104: what is meant by chaperone-export dependent transport? reply: We removed the word: “Chaperon”. (Line 112)
  10. Line 124-126: please rephrase the sentence. reply: The sentence is rephrased (Line 147/152)
  11. Line 127: Sentence is incomplete. reply: The sentence has been changed and integrated in the new paragraph (Line 147/152)
  12. Line 128: "In melanoma and TC, importin and exportin shuttle RAF proteins...": which importin and which exportin? reply: Based on our previuos studies, It read now: “In melanoma and TC, unidentified importin and CRM1 shuttle RAF proteins (particularly BRAF)” Line 122
  13. Line 149-151: Transition between the two sentences is unclear. reply: We modified the beginning of the sentence to be clear (Line 162)
  14. Line 166: "The nucleocytoplasmic transport of p53 is mediated by CRM1." Do you mean the nuclear export or do you mean that CRM1 regulates the localization of p53. CRM1 for sure does not mediate nuclear import of p53. reply: The sentence is corrected: “The nucleocytoplasmic export of p53” (Line 178)
  15. Line 172: "claudin-1 nuclear expression..": do you mean localisation? Or please specify. reply: “nuclear localization” instead of “nuclear expression” (Line 186)
  16. Line 180-183: please rephrase the sentence. reply: The sentence is rephrased (Line 194/196)
  17. Line 203: "Inhibiting importin and exportin...": which one? reply: The sentence is removed
  18. Line 203-205: the inhibitors do not inhibit the receptors "through phosphorylation of RanGDP to RanGTP". This is wrong in two respects: first, the inhibitors typically bind to the NES/NLS-binding pocket of the receptor, thereby preventing import/export complex formation. Second, the conversion of RanGDP to RanGTP does not occur by phosphorylation, but by an exchange. reply: The sentence is removed
  19. Line 226: "Swapping of DNA...": what do you mean by swapping? To me this is an inappropriate word to describe chromosomal translocations. reply: “The chromosomal translocation between” instead of swapping of DNA (Line 235)
  20. Line 257: "clinical trials with CRM1...": Do you mean LMB? reply:“Clinical trials with LMB” (Line 268)
  21. Line 259/260: the sentence is incomplete. reply: The sentence is completed now (Line 271)
  22. Line 280: "SINE inhibitors of nuclear import and export mechanisms...": SINEs do not inhibit nuclear import, as it is in their name: selective inhibitor of nuclear export, they target CRM1exclusively. reply: “SINE inhibitors of nuclear export” (Line 291)

Reviewer 2 Report

Dear Editors,

Thank you for consider me as reviewer, below please find my observations.

Review Report 1

Cells-1059954

“New insights into the link between melanoma and thyroid cancer: Role of nucleocytoplasmic trafficking”

The paper is a review being an interesting research, with an original contribution regarding the nucleocytoplasmic translocation pathways in thyroid cancer and malignant melanoma.

Major considerations:

  • Despite the fact that there are 124 references, the paper is a review that is not clearly if it is a systematic review or not; table 2 mention the major signal transducers mechanisms involved in both cancers, with references since 1999, the majority being quit old; I underline that the onco-genetic is a field where the major achievements occurred in the last years.
  • According to the available published data (ex. Vogt et al,2017 on behalf of ESMO) it is well known that multiple malignancies occur in 5-35%, especially in cancers where the survival is long. Thyroid cancer and mainly papillary thyroid cancer is a malignancy that has a prolonged survival rate; more than 90% of cases  are alive at 10 years. So, the occurrence of other malignancies is not a rare scenario. This fact should be mentioned, beyond the genetic mutations as common  
  • The authors mention in page 4/line 93:

Reference 40 do not represent an argument for this affirmation; this is a case report about a 61-year-old patient with malignant melanoma of foot stage IIB (not an advance one!) and a papillary thyroid cancer of left lobe, without any local and/or distant metastases, disease free at 24 months-irrelevant citation. Similarly for reference 41. There should be mention the link about the TSH receptors on the surface of melanocytes; also the link between the association of autoimmune thyroid disease (chronic lymphocytic thyroiditis) and malignant melanoma. Chronic lymphocytic thyroiditis is more frequently involved in the occurrence of thyroid cancer and is the major cause of indolent hypothyroidism.

-       There are no mention about the side effects of thyroid of immune therapy applied in advanced malignant melanoma. This is an important issue, due to the fact that also here are targeted commune genic pathways.

-       The conclusion section is too long; there is completely uncommon to have references cited in conclusions.

-       Moreover the authors have some conclusion that are not relied with the research.

Ex. “ treatment outcomes for different therapies for melanoma and TC vary based on drug target, cancer stage, and metastasis to distal sites”; please consider that this research has no analysis that sustain this sentence.

Author Response

Reviewer 2:

  1. reply:No, the review is not a systematic review and that is why it does not include all the references on the topic, which contains multiple aspects.
  2. Vogt et al 2017 (Multiple primary tumors: challenges and approaches, a review) is talking about multiple primary tumors and in our review we are focusing on the second primary tumor which is the consequences of the first primary tumor. If the survival rate of TC is prolonged, is not the case for the survival rate of the melanoma.
  3. We used the ref 40 as a support of our statement in Line 102 because, in the reference 40 abstract: “We suggest that the individuals who have cutaneous malignant melanoma may be predisposed to other primary cancers and especially thyroid carcinoma”. To address the concern of the respected reviewer, we added a sentence in the Line 104-107.
  4. We added a sentence talking about the relation of PDL1 blockade in melanoma and the endocrine side effect (thyroid dysfunction). Line 107-108
  5. We reduced the conclusion as advised by the reviewer
  6. We removed the sentence as suggested by the reviewer: “ treatment outcomes for different therapies for melanoma and TC vary based on drug target, cancer stage, and metastasis to distal sites”. The reference 123 was removed also.

Round 2

Reviewer 1 Report

The authors have largely addressed my previous comments and concerns. There are still some issues that need to be clarified prior to publication.

Line 65: "...transport of proteins, RNAs, signaling molecules and cargo molecules...": Here the same thing is said twice over, proteins, RNAs and signaling molecules are all cargo.

Line 131: a comma is missing after "respectively".

Line 151: "...which is an inhibitor of its import into the nucleus". It is not clear to what its is referring to: NF-kB or IkB.

Line 152: "..., causing its degradation and nuclear import...". Again not clear what its is referring to: IkB or the IKK complex.

Line 153 onwards: The second part of the sentence is not clear. Do you want to say that dysregulation of NF-kB in melanoma leads to resistance to BET inhibitors?

Line 155/156: What is meant by "unidentified importin"?

Line 158: proto-oncogenes Rat sarcoma (RAS) protein: singular or plural?

Line 166: Please use consistent writing: A-RAF, C-RAF versus BRAF.

Line 170: "...because it is associated with MEK1...": Do you want to say here that beta-catenin is associated with MEK1? Or that ERK2 or CRM1 is associated with it? Not clear.

Line 254: Here AKT is written in capital letters, before only with initial capital letter. Please be consistent.

Line 256: "beta-catenin (CTNNB1 gene) encoded on exon 3 activates ...": not clear.

Author Response

Line 65: "...transport of proteins, RNAs, signaling molecules and cargo molecules...": Here the same thing is said twice over, proteins, RNAs and signaling molecules are all cargo.

We removed: “and cargo molecules” Line 59

Line 131: a comma is missing after "respectively".

We added the coma Line 98

Line 151: "...which is an inhibitor of its import into the nucleus". It is not clear to what its is referring to: NF-kB or IkB.

We changed the sentence: “Which is an inhibitor of NF-kB import” Line118

Line 152: "..., causing its degradation and nuclear import...". Again not clear what its is referring to: IkB or the IKK complex.

“Causing IkB degradation” Line 119

Line 153 onwards: The second part of the sentence is not clear. Do you want to say that dysregulation of NF-kB in melanoma leads to resistance to BET inhibitors?

We made the sentence clearer: “ Dysregulation of NF-κB nucleocytoplasmic transport leads to the promotion of tumorigenesis in TC [14,44] and to BET inhibitors (Bromodomain and extra terminal protein inhibitors) resistance in melanoma” Lines 120-122

Line 155/156: What is meant by "unidentified importin"?

We still do not know which importin(s) are localizing RAFs to the nucleus. We already published on nuclear BRAF but we did not identify the transport mechanism yet…We are in the process to do so. 

Line 158: proto-oncogenes Rat sarcoma (RAS) protein: singular or plural?

“…by proto-oncogenes ­Rat sarcoma (RAS) proteins [46].” Line 126

Line 166: Please use consistent writing: A-RAF, C-RAF versus BRAF.

Done: ARAF, BRAF, CRAF         Lines 133-134

Line 170: "...because it is associated with MEK1...": Do you want to say here that beta-catenin is associated with MEK1? Or that ERK2 or CRM1 is associated with it? Not clear.

Modified sentence: “Nuclear export of ERK is inhibited by blocking CRM1 because the relocalization of nuclear ERK to the cytoplasm involves MEK1, which contains the NES sequence [57].” Line 135-137

Line 254: Here AKT is written in capital letters, before only with initial capital letter. Please be consistent.

All were checked: “Akt”

Line 256: "beta-catenin (CTNNB1 gene) encoded on exon 3 activates ...": not clear

Now: “… β-catenin (CTNNB1 gene) activates T-cell factor…” Line153

Reviewer 2 Report

The authors made an important improvement of the manuscript.

They responded and modified the manuscript  according to the comments addressed, showing a clear understanding of the requested issues.

Author Response

Thank you very much.